# MSAC-Net: 3D Multi-Scale Attention Convolutional Network for Multi-Spectral Imagery Pansharpening

Erlei Zhang [1], Yihao Fu [2], Jun Wang [2,3], Lu Liu [2], Kai Yu [2] and Jinye Peng [2,3,*]

1   School of Information Engineering, Northwest A&F University, Xi'an 712100, China;
    erlei.zhang@nwafu.edu.cn
2   School of Information Science and Technology, Northwest University, Xi'an 710127, China;
    201932142@stumail.nwu.edu.cn (Y.F.); jwang@nwu.edu.cn (J.W.); liulu@nwu.edu.cn (L.L.);
    yukai@nwu.edu.cn (K.Y.)
3   Shaanxi Province Silk Road Digital Protection and Inheritance of Cultural Heritage Collaborative Innovation
    Center, Xi'an 710127, China
*   Correspondence: pjy@nwu.edu.cn

**Abstract:** Pansharpening fuses spectral information from the multi-spectral image and spatial information from the panchromatic image, generating super-resolution multi-spectral images with high spatial resolution. In this paper, we proposed a novel 3D multi-scale attention convolutional network (MSAC-Net) based on the typical U-Net framework for multi-spectral imagery pansharpening. MSAC-Net is designed via 3D convolution, and the attention mechanism replaces the skip connection between the contraction and expansion pathways. Multiple pansharpening layers at the expansion pathway are designed to calculate the reconstruction results for preserving multi-scale spatial information. The MSAC-Net performance is verified on the IKONOS and QuickBird satellites' datasets, proving that MSAC-Net achieves comparable or superior performance to the state-of-the-art methods. Additionally, 2D and 3D convolution are compared, and the influences of the number of convolutions in the convolution block, the weight of multi-scale information, and the network's depth on the network performance are analyzed.

**Keywords:** deep learning; multi-spectral image; 3D convolutional; multi-scale cost

## 1. Introduction

Multi-spectral (MS) and panchromatic (PAN) images are two remote sensing image types acquired by optical satellites. While they often represent similar scenes, their spectral and spatial resolutions differ. PAN images have high spatial resolution (HR) but low spectral resolution, whereas MS images have high spectral resolution and low spatial resolution (LR). Pansharpening fuses LR-MS and HR-PAN, generating super-resolution MS images with high spatial resolution (HR-MS). This can provide higher quality remote sensing images for such as target detection [1–3], distribution estimation [4] and change detection [5–7].

The existing pansharpening techniques can be divided into four categories: component substitution [8,9], multi-resolution analysis [10,11], model-based optimization [12,13] and deep learning [14]. Component substitution-based methods achieve pansharpening by replacing MS images' partial spectral components with spatial information from PAN images. However, this technique causes spectral distortion [15]. Multi-resolution analysis methods [16,17] decompose the source and then synthesize the HR-MS images through fusion and inverse transform. Although this technique can maintain good spectral characteristics, it causes spatial distortion due to decomposition. Model-based optimization methods are limited by their dependence on appropriate prior knowledge and hyper-parameters.

In recent years, deep-learning-based research has achieved great success in image processing [18–24]. Masi et al. [18] were the first to propose a CNN-based pansharpening

method (PNN), whose structure is based on the super-resolution CNN [25]. Yuan et al. [19] designed a multi-scale and multi-depth CNN (MSDCNN), which introduces multi-scale information through two branches and a different number of learning blocks for deep feature learning. Yang et al. [26] proposed a residual network for pansharpening. However, most of the existing approaches focus on spatial feature extraction while paying less attention to the spectral information and spatial scale information of fusion images. Consequently, the fusion process is often characterized by spectral information loss or spatial feature redundancy.

The 3D convolution shows promise in volume data analysis [27–29]. For example, Mei et al. [30] used 3D CNN to extract spectral features of remote sensing super-resolution images. Compared to traditional 2D CNN methods, 3D CNN methods emphasize extracting spectral features while preserving spatial features [28]. Therefore, 3D convolution characteristics promote generating images with high spatial and spectral resolution.

Inspired by the human visual system [31], the attention mechanism was proved to have a positive effect on image understanding [32] and has been widely applied in image processing due to its focus on local information [33]. For example, Wang et al. [34] presented a model with several attention blocks and combined these with the residual structure to enable focusing specific areas while reducing the number of calculations. Mei et al. [32] utilized attention mechanisms to study spatial and spectral correlations between adjacent pixels. U-Net (Figure 1) [35], named by its U-shaped structure, has a high capability to extract and represent features and combine low- and high-scale semantic information for semantic segmentation of input images. Guo et al. [36] replaced the basic convolution block of the decoder in the U-Net architecture with the residual channel attention block [37], consequently improving the model capability. Oktay et al. [38] proposed the attention gate (AG) module based on the U-Net architecture to enable automated focus on target structures of different shapes and sizes while suppressing irrelevant areas.

At present, most MS pansharpening methods focus on the injection of spatial details, but ignore the multi-scale features and spectral features of multi-spectral images. This work proposes a novel 3D multi-scale attention deep convolutional network (MSAC-Net) for MS imagery pansharpening. Following the U-Net framework, MSAC-Net consists of a contraction path that extracts high-scale features from LR-MS and HR-PAN (Figure 2, left) and an expansion path that fuses spatial and spectral information (Figure 2, right).The attention mechanism is introduced in the contraction and expansion paths instead of skip connections, supporting focusing spatial details of the feature maps. Furthermore, the deep supervision mechanism enables MSAC-Net to utilize multi-scale spatial information by adding a pansharpening layer at each scale of the expansion path. The results demonstrate that MSAC-Net achieves high performance in both spatial and spectral dimensions. In summary, this work:

1. Designs a 3D CNN to probe the spectral correlation of adjacent band images, thus reducing the spectral distortion in MS pansharpening;
2. Uses a deep supervision mechanism that utilizes multi-scale spatial information to solve the spatial detail missing problem;
3. Applies the AG mechanism instead of the skip connection in U-Net structure and presents experiments demonstrating its advantages in MS pansharpening.

The rest of the manuscript is organized as follows. Section 2 presents the related work, while Section 3 introduces the proposed MSAC-Net. Section 4 describes and analyzes the experimental results. Finally, Section 5 concludes the paper with a short overview of the contributions.

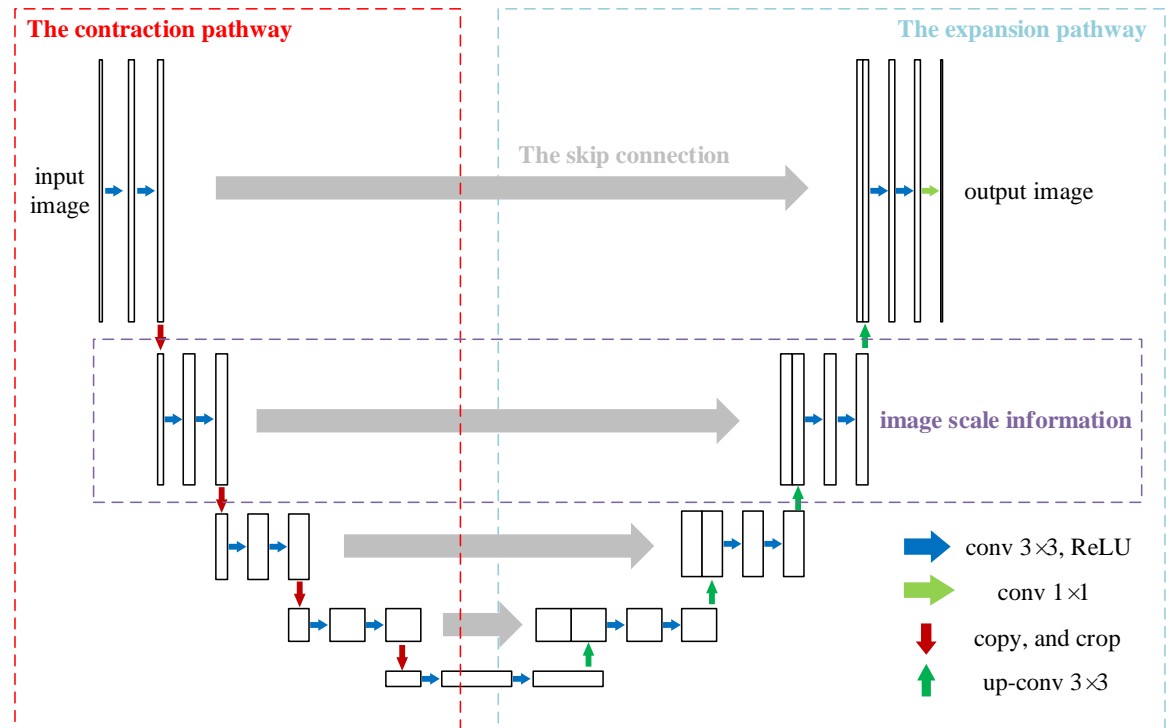

**Figure 1.** Classic U-Net design.

**Figure 2.** The proposed MSAC-Net architecture.

## 2. Related Work

Ronneberger et al. [35] first proposed the U-Net network for medical image segmentation. Because its structure is to combine low-level and high-level semantic information for semantic segmentation of the input image, U-Net has high feature representation and extraction ability. Therefore, in this section, the various components of U-Net and improvements based on U-Net are introduced.

According to the difference of feature information extracted by U-Net, its composition can be divided into three parts.

1. The contraction pathway (i.e., the encoder). It is mainly used for shallow feature extraction and coding. For the improvement of this component, Milletari et al. [39] referred to residual learning for the design of each convolution block on the basis of U-Net, which makes V-Net learn image features more fully. Wang et al. [40] used ResNet-50 pre-trained on the ImageNet dataset as its encoder to accelerate the convergence and achieve better performance via transfer learning.

2. The expansion pathway (i.e., the decoder). The purpose of this design is to restore or enlarge the size of feature map. Therefore, the design directly affects the final result of feature recovery. Guo et al. [36] learned an additional dual regression mapping after expansion pathway to estimate the down-sampling kernel and reconstruct LR images, which forms a closed-loop to provide additional supervision. Banerjee et al. [41] designed decoder-side dense skip pathways, making the features under the final scale contain the feature results of all scales. In addition, Khalel et al. [42] realized multi-task learning with the design of dual encoder. Ni et al. [43] extracted and fused the pre-output features and deep features of the network again, thus realizing the multi-task learning of the network.

3. The skip connection. The skip connections transfer the shallow feature maps of the network to the deep layer, which is helpful for the training of the deep network. Zhou et al. [44] proposed U-Net++, in which each scale feature is designed to carry out feature transfer to a larger scale, thus forming a dense connection structure at skip connections. Ni et al. [43] also added sparable gate unit at the skip connection to improve the accuracy of image spatial feature extraction. Rundo et al. [45] introduced squeeze-and-excitation blocks instead of the skip connection to expect an increased representational power from modeling the channel-wise dependencies of convolutional features [46]. Wang et al. [47] designed a module to generate attentional feature maps by paying attention to the *H*, *W* and *C* of feature maps to replace the skip connection.

In the U-Net, feature maps are learned by "compression-expansion". In addition, there are studies targeting U-Net as a whole for improvement. Yang et al. [48] transformed U-Net as a spatial attention module and inserted it into the network as a branch to extract spatial features of images. Wei et al. [49] replaced each convolution block in the U-Net structure with a small U-Net structure to achieve multi-scale feature extraction. Xiao et al. [50] adopted a dual U-Net structure to inject features extracted from the external U-Net into the internal U-Net, achieving the effect of multi-stage detail injection.

## 3. Method

This section will introduce the overall design of MSAC-Net, including its structure, AG module and in-depth monitoring mechanism.

First, we will introduce the pansharpening model based on CNN. Let the LR-MS image with the size $h \times w \times c$ be denoted $M_{LR} \in R^{h \times w \times c}$, Similarly, denote the HR-PAN with $H \times W$ size as $P_{HR} \in R^{H \times W}$ and HR-MS as $M_{HR} \in R^{H \times W \times c}$. Taking LR-MS and HR-PAN as inputs, the pansharpening task of generating the HR-MS can be expressed as:

$$\ell(\theta) = \|\mathcal{M}(M_{LR}, P_{HR}; \theta) - M_{HR}\| \tag{1}$$

where $\mathcal{M}(\cdot)$ represents the mapping from the CNN's input to the out, $\theta$ denotes parameters to be optimized and $\| \cdot \|$ is a loss function. The CNN can learn the involved knowledge from input data, offering the possibility for MS pansharpening. Table 1 shows the de-

sign of all modules in MSAC-Net.The following subsections introduce the details on the representation of these elements within MSAC-Net.

**Table 1.** The design of the MSAC-Net.

| Module | Input Size | Layer (Filter Size) or Method | Filters | Output Size |
|---|---|---|---|---|
| Down$_1$ | $2 \times c \times H_1 \times W_1$ <br> $64 \times c \times H_1 \times W_1$ | $(Conv3D(3,3,3), ReLU) \times 2$ <br> $Maxpool(1,2,2)$ | 64 <br> – | $64 \times c \times H_1 \times W_1$ <br> $64 \times c \times H_2 \times W_2$ |
| Down$_2$ | $64 \times c \times H_2 \times W_2$ <br> $128 \times c \times H_2 \times W_2$ | $(Conv3D(3,3,3), ReLU) \times 2$ <br> $Maxpool(1,2,2)$ | 128 <br> – | $128 \times c \times H_2 \times W_2$ <br> $128 \times c \times H_3 \times W_3$ |
| Down$_3$ | $128 \times c \times H_3 \times W_3$ <br> $256 \times c \times H_3 \times W_3$ | $(Conv3D(3,3,3), ReLU) \times 2$ <br> $Maxpool(1,2,2)$ | 256 <br> – | $256 \times c \times H_3 \times W_3$ <br> $256 \times c \times H_4 \times W_4$ |
| Down$_4$ | $256 \times c \times H_4 \times W_4$ <br> $512 \times c \times H_4 \times W_4$ | $(Conv3D(3,3,3), ReLU) \times 2$ <br> $Maxpool(1,2,2)$ | 512 <br> – | $512 \times c \times H_4 \times W_4$ <br> $512 \times c \times H_5 \times W_5$ |
| Down$_5$ | $512 \times c \times H_5 \times W_5$ <br> $1024 \times c \times H_5 \times W_5$ | $(Conv3D(3,3,3), ReLU) \times 2$ <br> $Tran([1,4,4], stride = [1,2,2])$ | 1024 <br> 512 | $1024 \times c \times H_5 \times W_5$ <br> $512 \times c \times H_4 \times W_4$ |
| Up$_5$ | $512 \times c \times H_4 \times W_4 \oplus AG_5$ <br> $512 \times c \times H_4 \times W_4$ | $(Conv3D(3,3,3), ReLU) \times 2$ <br> $Tran([1,4,4], stride = [1,2,2])$ | 512 <br> 256 | $512 \times c \times H_4 \times W_4$ <br> $256 \times c \times H_3 \times W_3$ |
| Up$_4$ | $256 \times c \times H_3 \times W_3 \oplus AG_4$ <br> $256 \times c \times H_3 \times W_3$ | $(Conv3D(3,3,3), ReLU) \times 2$ <br> $Tran([1,4,4], stride = [1,2,2])$ | 256 <br> 128 | $256 \times c \times H_3 \times W_3$ <br> $128 \times c \times H_2 \times W_2$ |
| Up$_3$ | $128 \times c \times H_2 \times W_2 \oplus AG_3$ <br> $128 \times c \times H_2 \times W_2$ | $(Conv3D(3,3,3), ReLU) \times 2$ <br> $Tran([1,4,4], stride = [1,2,2])$ | 128 <br> 64 | $128 \times c \times H_2 \times W_2$ <br> $64 \times c \times H_1 \times W_1$ |
| Up$_2$ | $64 \times c \times H_1 \times W_1 \oplus AG_2$ | $(Conv3D(3,3,3), ReLU) \times 2$ | 64 | $64 \times c \times H_1 \times W_1$ |
| Output | $64 \times c \times H_4 \times W_4$ | $Conv3D(1,1,1)$ | 1 | $1 \times c \times H_1 \times W_1$ |
| Reconstruction | $F_i \times c \times H_i \times W_i$ | $Conv3D(1,1,1)$ | 1 | $1 \times c \times H_i \times W_i$ |
| AG$_{i+1}$ | $(Down_i)F_i \times c \times H_i \times W_i$ <br> $(Up_{i+1})F_i \times c \times H_i \times W_i$ | $Conv3D(1,1,1)$ | $F_i/2$ | $(x)F_i/2 \times c \times H_i \times W_i$ <br> $(g)F_i/2 \times c \times H_i \times W_i$ |
| | $(x)F_i/2 \times c \times H_i \times W_i$ <br> $(g)F_i/2 \times c \times H_i \times W_i$ | $ReLU(x+g))$ | – | $F_i \times c \times H_i \times W_i$ |
| | $F_i \times c \times H_i \times W_i$ | $Conv3D(1,1,1) + sigmoid$ | 1 | $1 \times c \times H_i \times W_i$ |
| | $1 \times c \times H_i \times W_i$ <br> $(Down_i)F_i \times c \times H_i \times W_i$ | $\otimes$ | – | $F_i \times c \times H_i \times W_i$ |

### 3.1. The MSAC-Net's Structure

As shown in Figure 2, MSAC-Net consists of three parts: data preprocessing, the contraction pathway (left sub-network) and the expansion pathway (right sub-network). The contraction pathway and the expansion pathway have the same effect as U-Net. Placed between the two paths, the AG mechanism replaces the skip connection, thus improving the local detail feature representation ability. Moreover, MSAC-Net uses multi-scale information pansharpening layers to resolve the MS pansharpening problems.

The traditional 2D CNN approaches commonly cascade $P_{HR}$ with the band dimension of $M_{LR}$ to obtain the input data $X \in R^{H \times W \times (c+1)}$. In contrast, the proposed MSAC-Net uses 3D cube data as input. As shown in Figure 2, data preprocessing converts the PAN image $P_{HR}$ to $P'_{HR} \in R^{H \times W \times c}$, where $P'_{HR_i} = P_{HR}, P'_{HR_i} \in \{i | i \in c, R^{H \times W \times c}\}$. Similarly, the interpolation algorithm converts $M_{LR}$ to $M'_{HR} \in R^{H \times W \times c}$. Finally, the input image $X \in R^{2 \times H \times W \times c}$ is obtained by $R(P'_{HR} \oplus M'_{HR})$, where $\oplus$ is the cascade operation, $R(\cdot)$ represents the resizing operation and the number "2" stems from the two features, MS and PAN.

In the contraction pathway, the $i$-th low-scale feature $F_{L_i}$ is obtained as:

$$F_{L_i} = Block(Max(F_{L_{i-1}})) \tag{2}$$

where $Block(\cdot)$ is is composed of two groups of kernels with a rectified linear unit (ReLU) as an activation function, (i.e., $3 \times 3 \times 3$ kernel + ReLU), where $Max(\cdot)$ denotes the max-pooling layer with a $1 \times 2 \times 2$ kernel. The $1 \times 2 \times 2$ kernel is chosen to down-sampled $H$ and $W$ with a factor of two while keeping the number of bands ($c$) unchanged.

In the expansion pathway, the *i*-th high-scale feature $F_{H_i}$ is obtained as follows:

$$F_{H_i} = Tran(Block(F_{H_{i+1}} \oplus AG(F_{L_{i+1}}, F_{H_{i+1}}))) \tag{3}$$

where $Tran(\cdot)$ represents the transposed convolution with a factor of two, and $AG(\cdot)$ represents stands for the AG module. The pansharpening layer is set after each scale of the convolution block. The next sections introduce further details on the AG module and the pansharpening layer will be introduced in a later section.

### 3.2. The Attention Gate (AG) Module

Figure 3 depicts the details of the AG module's structure. The inputs on the *i*-th AG module are $F_{H_i} \in R^{F_i \times c \times H_i \times W_i}$ and $F_{L_i} \in R^{F_i \times c \times H_i \times W_i}$. First, the $F_{H_i}$ and $F_{L_i}$ change the numbers of channels from $F_i$ to $F_i/2$ using a $3 \times 3 \times 3$ kernel and are then cascade. Next, the gate feature ($q_{att}^i$) is obtained via the ReLU function and $1 \times 1 \times 1$ kernel. Finally, $q_{att}^i$ is activated using a sigmoid function and multiplied by $F_{L_i}$ to obtain the feature $\hat{x}_{up_i}$. In contrast to [37], this work uses gate features instead of global pooling because the extracted gate features are more consistent with the PAN features. The AG module's operations can be formalized as:

$$q_{att}^i = Conv(\sigma_1(F_{L_i} \oplus F_{H_i}); \theta) \tag{4}$$

$$\hat{x}_{up_i} = \sigma_2(q_{att}^i) \otimes F_{L_i} \tag{5}$$

where $\sigma_1$ is the ReLU activate function, $\sigma_2$ denotes the sigmoid activation, $\theta$ is a parameter and $\otimes$ stands for pixel-by-pixel multiplication on each feature map. Finally, $F_{H_{i-1}}$ is obtained by the convolution block using the transposed convolution.

$$F_{H_{i-1}} = Tran(Block(\hat{x}_{up_i})) \tag{6}$$

Utilizing the AG module instead of the skip connection enables MSAC-Net to pay more attention to each scale's local spatial details, consequently improving the spatial performance of pansharpening.

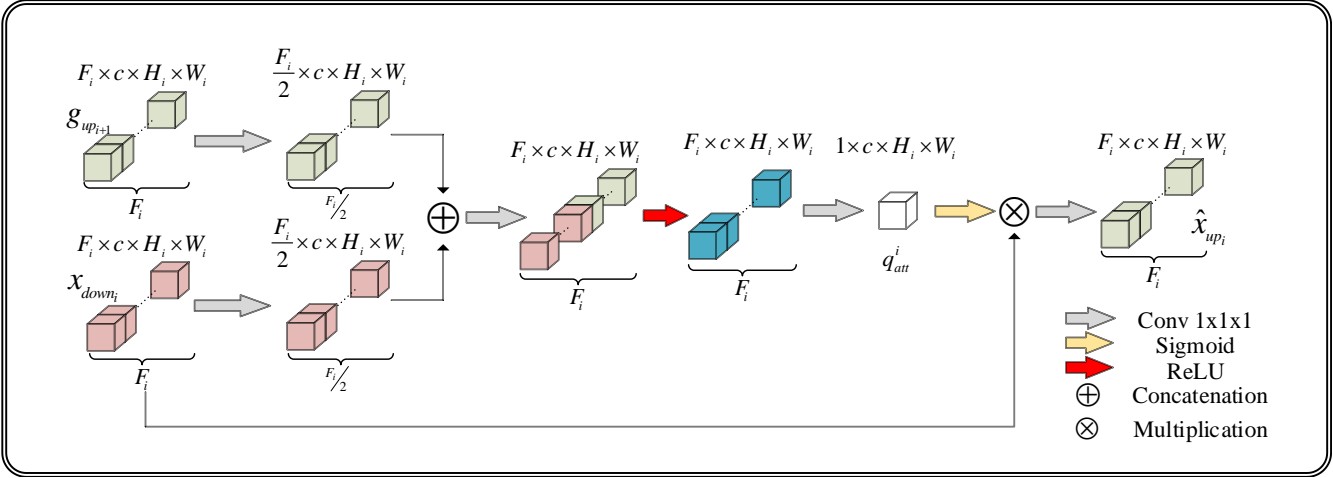

**Figure 3.** The attention gate (AG) module.

### 3.3. Pansharpening Layer and Multi-Scale Cost Function

MSAC-Net has $S$ scales in the expansion pathway, i.e., $S$ scale spaces. The pansharpening layer at each scale reconstructs the high-scale feature using a $1 \times 1 \times 1$ convolution kernel, obtaining $\hat{Y}_{s_i}$. The formula is:

$$\hat{Y}_{s_i} = R(Block(\hat{x}_{up_i})) \tag{7}$$

where $R(\cdot)$ is the $i$-th scale pansharpening layer. Accordingly, $Y_{s_i}$ is generated using the bicubic interpolation, given as:

$$Y_{s_i} = D(Y_{s_{i-1}}) \tag{8}$$

where $Y_{s_1}$ represents the ground truth and $D(\cdot)$ is the bicubic interpolation with a factor of 2.

The $\ell_1$-norm loss is used to constrain $Y_{s_i}$ and $\hat{Y}_{s_i}$ at the $i$th scale. The formula is expressed as:

$$\ell_{s_i} = \|\hat{Y}_{s_i} - Y_{s_i}\|_1 \tag{9}$$

where $\ell_{s_i}$ denotes the loss at the $i$-th scale.

Finally, the proposed MSAC-Net's multi-scale cost function is calculated as (see Figure 2):

$$
\begin{aligned}
L &= \ell_1 + \lambda \sum_{i=2}^{S} \ell_{s_i} \\
&= \|\hat{Y}_{s_1} - Y_{s_1}\|_1 + \lambda \sum_{i=2}^{S} \|\hat{Y}_{s_i} - Y_{s_i}\|_1
\end{aligned}
\tag{10}
$$

where $\lambda$ is the weight of multi-scale information.

## 4. Results

### 4.1. Experimental Setup

4.1.1. Datasets & Parameter Settings

To test the effectiveness of MSAC-Net, we used datasets collected by IKONOS (http://carterraonline.spaceimaging.com/cgi-bin/Carterra/phtml/login.phtml, accessed on 20 December 2018) and QuickBird (http://www.digitalglobe.com/product-samples, accessed on 20 December 2018) satellites (Figure 4). Both datasets contain four standard colors (R: red, G: green, B:blue, N:near infrared). In order to ensure the availability of the ground truth, Wald's protocol [51] was used to obtain baseline images in training and simulation tests.

The steps of obtaining simulation data are as follows:

(1) The original HR-PAN and LR-MS images were down-sampled with a factor of 4;
(2) The down-sampled HR-PAN was used as the input PAN, and the down-sampled LR-MS was used as the input LR-MS;
(3) The original LR-MS was used as ground truth in the simulation experiment.

In the real datasets, we only normalized the original image. Therefore, the fusion image obtained from real datasets does not have ground truth.

The dataset information is shown in Table 2. The selected dataset contains rich texture information, such as rivers, roads, mountains, etc. We cropped images randomly from each dataset to 3000 PAN/MS data pairs for a total of 6000 data pairs. The PAN size of $256 \times 256 \times 1$ and MS size of $64 \times 64 \times 4$. Empirically, all data pairs were divided into a training set, validation set and test set in a ratio of 6:3:1. In the experiment of the real dataset, 50 data pairs with PAN image size of $1024 \times 1024 \times 1$ and MS image size of $256 \times 256 \times 4$ were randomly acquired for each dataset.

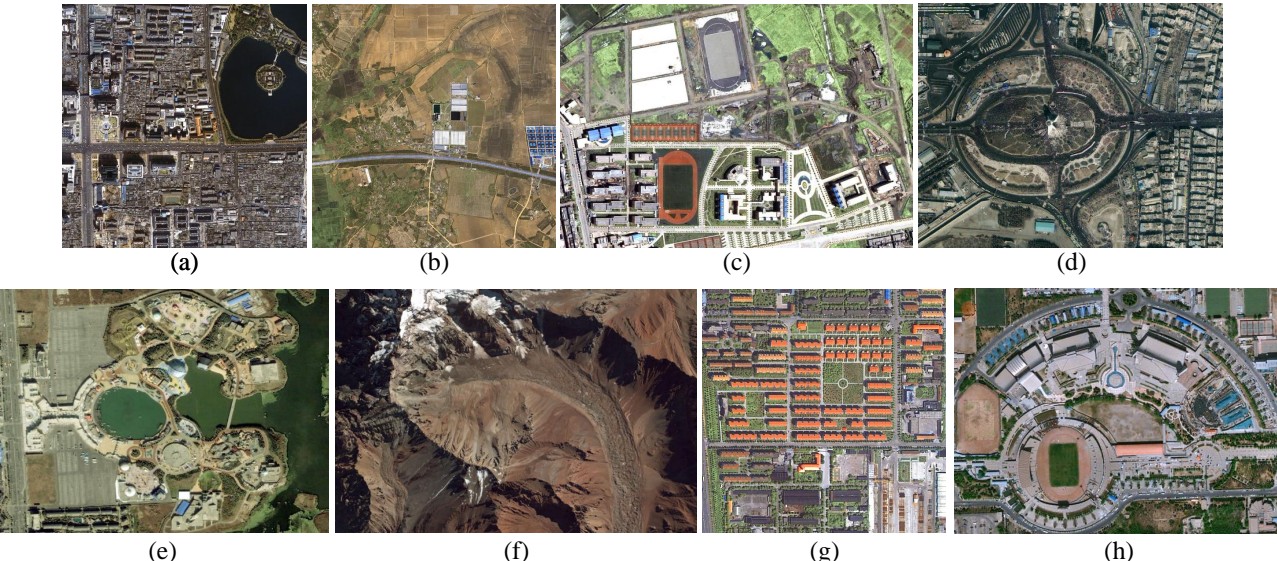

**Figure 4.** Display images in datasets. (**a–d**) are the IKONOS dataset's images. (**e–h**) are the QuickBird dataset's images.

**Table 2.** Dataset information.

| Satellite | Pan | Blue | Green | Red | NIR | Resolution (PAN/MS) |
|---|---|---|---|---|---|---|
| IKONOS | 450–900 | 450–530 | 520–610 | 640–720 | 760–860 | 1 m/4 m |
| QuickBird | 450–900 | 450–520 | 520–600 | 630–690 | 760–900 | 0.7 m/2.8 m |

The proposed method was implemented in Python 3.6 using the Pytorch-1.7 framework and trained and tested on an NVIDIA 1080 GPU. The stochastic gradient descent algorithm was used to converge during training; its parameter settings are shown in Table 3. Among them, the learning rate decay was halved every 2000 iterations. During the training stage, we saved the best model, which achieved the best performance on the validation dataset, and used it for the test.

**Table 3.** Parameter list.

| Learning Rate | Weight Decay | Momentum | Batch Size | Iteration |
|---|---|---|---|---|
| 0.01 | $10^{-3}$ | 0.9 | 16 | $2 \times 10^4$ |

#### 4.1.2. Compared Methods

The proposed method is validated through a comparison with several pansharpening methods, including GS [52], Indusion [10], SR [13], PNN [18], PanNet [26], MSDCNN [19], MIPSM [20] and GTP-PNet [24]. GS is based on component substitution. Indusion is a multi-resolution analysis method. Furthermore, SR is based on sparse representation learning technology, whereas PNN is based on a three-layered CNN. PanNet is a residual network based on high-pass filtering. MSDCNN introduces residual learning and constructs a multi-scale and multi-depth feature extraction based on CNN. MIPSM is a CNN fusion model that uses dual branches to extract features. Lastly, GTP-PNet is a residual learning network based on gradient transformation prior. PNN, PanNet, MSDCNN, MIPSM and GTP-PNet are the most advanced deep learning methods presented in recent literature.

#### 4.1.3. Performance Metrics

The performance of the proposed pansharpening method is analyzed through quantitative and visual assessments. We selected six reference evaluation indicators and one

non-reference evaluation indicator. All symbols in the reference evaluation indicators are explained in Table 4, and the reference evaluation indicators are as follows:

(1)  The correlation coefficient (*CC*) [53]: *CC* reflects the similarity of spectral features between the fused image and the ground truth. $CC \in [0,1]$, with 1 being the best attainable value. *CC* can be expressed as follows:

$$CC(\hat{X}, X) = \frac{1}{M} \sum_{m=1}^{M} \frac{\sum_{n=1}^{N} (\hat{X}_n^m - \bar{\hat{X}})(X_n^m - \bar{X})}{\sqrt{\sum_{n=1}^{N} (\hat{X}_n^m - \bar{\hat{X}})^2 (X_n^m - \bar{X})^2}} \qquad (11)$$

(2)  Peak signal-to-noise ratio (*PSNR*) [54]: *PSNR* is an objective measure of the information contained in an image. A larger value demonstrates that there is less distortion between the two images. *PSNR* can be expressed as follows:

$$PSNR(\hat{X}, X) = 10 \lg \left( \frac{L^2}{MSE(\hat{X}, X)} \right) \qquad (12)$$

(3)  Spectral angle mapper (*SAM*) [55]: *SAM* calculates the overall spectral distortion between the fused image and the ground truth. $SAM \in [0,1]$, with 0 being the best attainable value, is defined as follows:

$$SAM(\hat{X}, X) = \frac{1}{N} \sum_{n=1}^{N} \arccos \left( \frac{\langle \hat{x}_n, x_n \rangle}{\|\hat{x}_n\| \|x_n\|} \right) \qquad (13)$$

(4)  Root mean square error (*RMSE*) [56]: *RMSE* measures the deviation between the fused image and the ground truth. $RMSE \in [0,1]$, with 0 being the best attainable value, is defined as follows:

$$RMSE(\hat{X}, X) = \sqrt{MSE(\hat{X}, X)} \qquad (14)$$

(5)  *Erreur relative globale adimensionnelle de synthèse* (*ERGAS*) [57]: *ERGAS* represents the difference between the fused image and the ground truth. $ERGAS \in [0,1]$, with 0 being the best attainable value, can be expressed as follows:

$$ERGAS(\hat{X}, X) = 100d \sqrt{\frac{1}{M} \sum_{m=1}^{M} \left( \frac{RMSE(\hat{X}_m, X_m)}{\bar{X}_m} \right)^2} \qquad (15)$$

(6)  Structural similarity index measurement (*SSIM*) [54]: *SSIM* measures the similarity between the fusion image and the ground truth image. $SSIM \in [0,1]$, with 1 being the best attainable value, is defined as follows:

$$SSIM(\hat{X}, X) = \left( \frac{2\bar{\hat{X}}\bar{X} + C_1}{\bar{\hat{X}}^2 + \bar{X}^2 + C_1} \right)^{\alpha} \left( \frac{2\sigma_{\hat{X}}\sigma_X + C_2}{\sigma_{\hat{X}}^2 + \sigma_X^2 + C_2} \right)^{\beta} \left( \frac{\sigma_{\hat{X}X} + C_3}{\sigma_{\hat{X}}\sigma_X + C_3} \right)^{\gamma} \qquad (16)$$

(7)  Quality without reference (*QNR*) [58]: As a non-reference evaluation indicator, *QNR* compares the brightness, contrast and local correlation of the fused image with the original image. $QNR \in [0,1]$, with 1 being the best attainable value, is defined as follows:

$$QNR = (1 - D_\lambda)^a (1 - D_S)^b \qquad (17)$$

where usually $a = b = 1$ and the spatial distortion index $D_s$ and the spectral distortion index $D_\lambda$ are based on universal image quality index ($Q$) [59]. Furthermore, $D_s, D_\lambda \in [0, 1]$, with 0 being the best attainable value. $Q$ is defined as:

$$Q(\hat{X}, X) = \frac{\sigma_{\hat{X}X}}{\sigma_{\hat{X}}\sigma_X} \times \frac{2\sigma_{\hat{X}}\sigma_X}{\sigma_{\hat{X}}^2 + \sigma_X^2} \times \frac{2\bar{\hat{X}}\bar{X}}{\bar{\hat{X}}^2 + \bar{X}^2} \tag{18}$$

thus, $D_s$ and $D_\lambda$ are defined as:

$$D_s = \sqrt[p]{\frac{1}{M} \sum_{m=1}^{M} \left| Q(\hat{X}_m, PAN) - Q(X_m, PAN_L) \right|^p} \tag{19}$$

$$D_\lambda = \sqrt[q]{\frac{1}{M(M-1)} \sum_{m=1}^{M} \sum_{\substack{r=1 \\ r \neq m}}^{M} \left| Q(\hat{X}_m, \hat{X}_r) - Q(MS_L^m, MS_L^r) \right|^q} \tag{20}$$

**Table 4.** Formula symbol table.

| Symbol | Means | Symbol | Means |
|--------|-------|--------|-------|
| $\hat{X}$ | Input image | $X$ | Ground Truth |
| $\bar{\hat{X}}$ | Sample means of $\hat{X}$ | $\bar{X}$ | Sample means of $X$ |
| $\hat{x}$ | A pixel in $\hat{X}$ | $x$ | A pixel in $X$ |
| $N$ | Number of pixels | $M$ | Number of band |
| $MSE()$ | Mean square error | $L$ | Gray level |
| $\alpha$ | Brightness parameter | $\beta$ | Contrast parameter |
| $\gamma$ | Structural parameter | $C_i$ | Infinitely small constant |
| $\sigma_{\hat{X}}$ | Variances of $\hat{X}$ | $\sigma_X$ | Variances of $X$ |
| $\sigma_{\hat{X}X}$ | Covariance of $\hat{X}$ and $X$ | $p, q$ | Positive integer |
| $d$ | PAN image spatial resolution/MS image spatial resolution | | |

### 4.2. The Influences of Multi-Scale Fusion and Attention Gate Mechanism

This section discusses MSAC-Net's and other methods' results and analyzes the benefits of each MSAC-Net module. Considering that the image has the characteristics of scale invariance and U-Net will scale the image at multiple scales in the learning process, MSAC-Net designed a deep supervision mechanism to restrain the multi-scale fusion process. The design of the deep supervision mechanism is to ensure that in the fusion process of each scale, its features are constrained to ensure that the main features of the image are learned by the network. The AG mechanism can gradually suppress the feature regions of unrelated regions, thus highlighting the feature regions [38]. For image fusion, because MS images contain a lot of noise [60], MSAC-Net suppressed the noise area through the AG mechanism, to highlight the feature area. In addition, MSAC-Net uses 3D convolution to preserve the spectral information of MS data and injects the PAN images' spatial information into MS images. These settings allow extracting better performance indicators while retaining the spectral correlation between bands.

#### 4.2.1. The Influence of the Pansharpening Layer

Figure 5 visualizes the reconstructed images Figure 5f–j and reference images Figure 5a–e at each scale. It can be observed that as the scale increases, the details of the reconstructed images Figure 5f–j become more similar to the intermediate reference images Figure 5a–e, indicating that the details of $\hat{Y}_{s_i}$ match closer the details of $Y_{s_i}$ at each scale. These findings demonstrate that adding the multi-scale cost function enables MSAC-Net to learn the features of each scale easily.

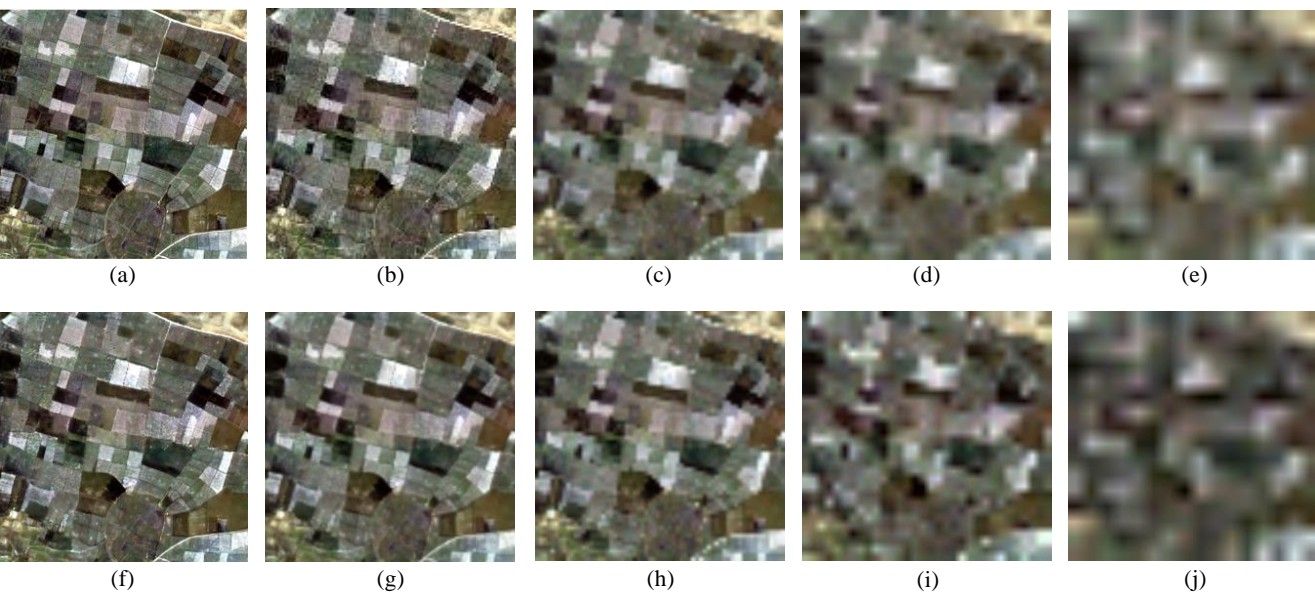

**Figure 5.** The reference image and pansharpening image are compared at different scales. The first line is the reference image for each scale, and the second line is pansharpening image for each scale. (**a**–**e**) are the reference images of the first to fifth layers respectively, and corresponding, (**f**–**j**) are the reconstructed images of the first to fifth layers respectively.

### 4.2.2. The Influence of the AG Module

In Figure 6, the feature maps of the first scale with and without the AG module are visualized. More precisely, Figure 6b,e are represent the feature maps without the AG module, and Figure 6c,f show the feature maps with the AG module. One can notice that the details in Figure 6c are significantly enhanced compared to those in Figure 6b. The detailed features regarding "rivers" and "land" in Figure 6c are extracted. Figure 6e has lower contrast and blurry edges, whereas Figure 6f has sharper contrast and sharper edges. Compared with $F_{L_i}$, $F_{H_i}$ contains more local spatial information, and the texture information is enhanced, yielding more accurate results. Therefore, these results demonstrates the AG module's utility in advancing the MSAC-Net's learning of detailed spatial features.

### 4.2.3. The Influence of Different Structures

Figure 7 compares the pansharpening results of four methods with different structures.

From the perspective of image visual perception, Figure 7b–e do not differ significantly. As can be observed from Figure 6 and Table 5, the spatial details can be extracted more fully by adding the AG mechanism, albeit with a slight loss of the spectral information. In Table 5, SAM of the "U-Net + scale" method is significantly reduced, and SSIM and PSNR are slightly improved when multi-scale information is introduced. Compared with Figure 7f, the spatial detail error of Figure 7h is reduced. These results show that the reconstruction of multi-scale information can make use of the U-Net's hierarchical relationships to improve the network's expression ability. As compared with Figure 7f–h, Figure 7i shows the smallest difference between the reference image and the method's output. Table 5 demonstrates the MSAC-Net's superiority stemming from its use of the AG module and multi-scale information.

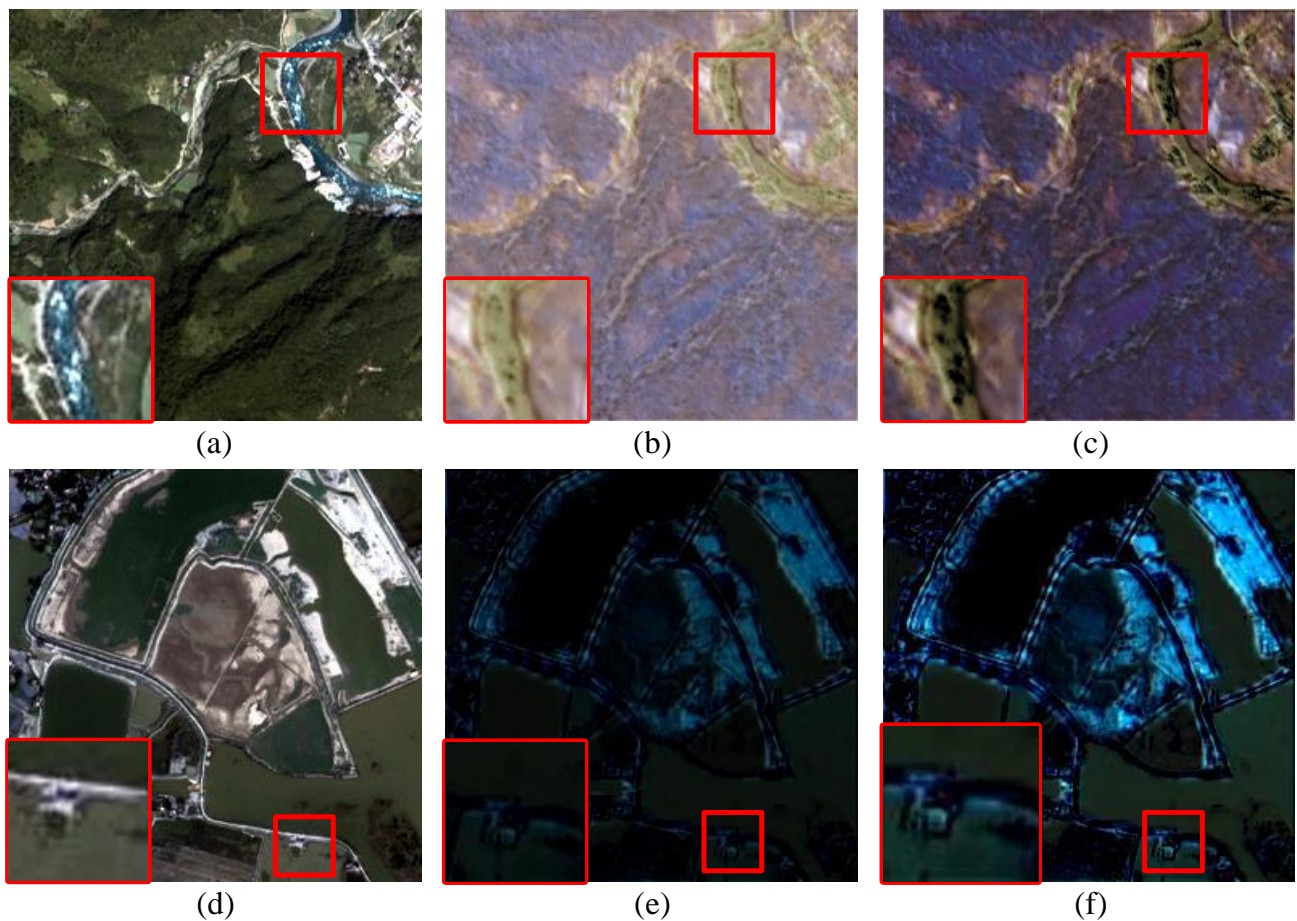

**Figure 6.** The comparison of feature maps with and without the AG module. (**a**) The ground truth. (**b**) The 53rd feature map in $F_{L_1}$. (**c**) The 53rd feature map in $F_{H_1}$. (**d**) the ground truth. (**e**) The 23rd feature map in $F_{L_1}$. (**f**) The 23rd feature map in $F_{H_1}$.

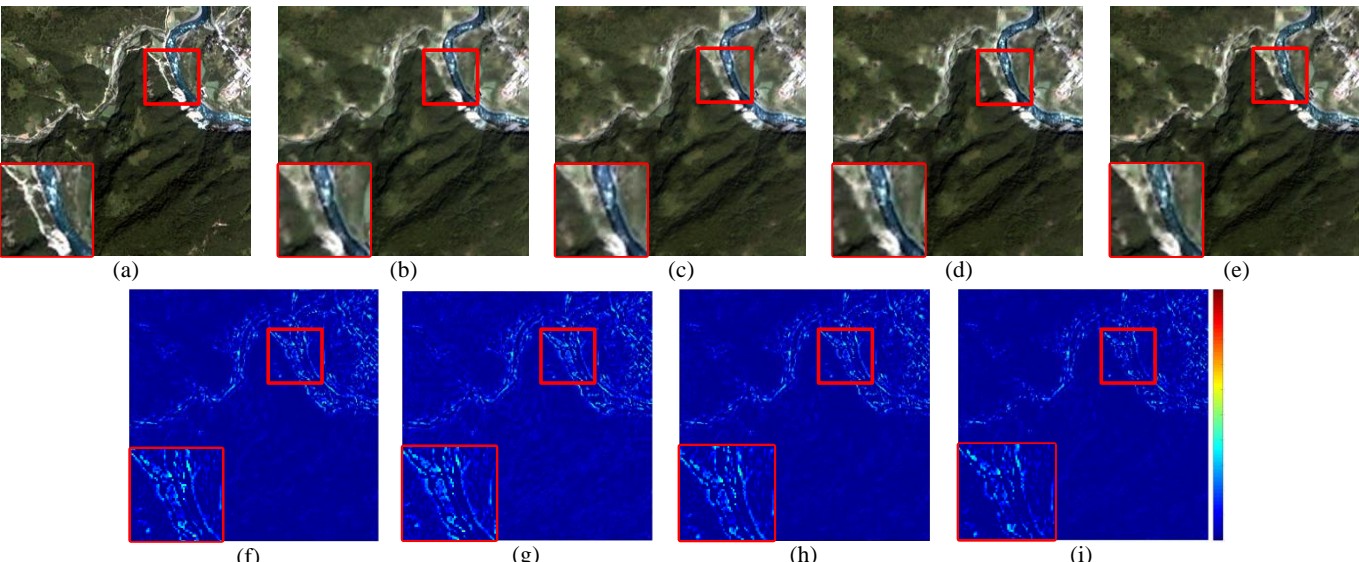

**Figure 7.** The comparison of the results for different structures. (**a**) The ground truth. (**b**) U-net. (**c**) U-net + AG: U-net with AG on the skip connections. (**d**) U-net + scale: U-Net with multi-scale cost function. (**e**) MSAC-Net. (**f**–**i**) are the spectral distortion maps corresponding to (**b**–**e**).

**Table 5.** The comparison of the results for different structures on IKONOS.

| Structures | CC | PSNR | SAM | ERGAS | RMSE | SSIM |
|---|---|---|---|---|---|---|
| U-Net | 0.8959 | 21.5200 | 5.1670 | 7.8263 | 0.0839 | 0.6525 |
| U-Net + AG | 0.8949 | 21.5247 | 5.2721 | 7.8407 | 0.0839 | 0.6543 |
| U-Net + scale | 0.8955 | 21.5754 | 4.8187 | 7.6403 | 0.0837 | 0.6624 |
| Our | **0.8968** | **21.6184** | **4.4281** | **7.4303** | **0.0830** | **0.6755** |

**Bold** means the best results.

### 4.3. Comparison of 2D and 3D Convolutional Networks

Within this subsection, 2D and 3D convolution are used in MSAC-Net, and their indicators are compared to assess the advantages of employing 3D convolution in MS imagery pansharpening. The experimental results are shown in Figure 8 and Table 6.

The visual assessment indicates that Figure 8c has significant spatial distortion, whereas Figure 8d is more similar to the ground truth. For example, the road in Figure 8d is clearer than in Figure 8c. The reason may lay in the fact that 3D convolution considers the spectral characteristics of the MS image, thus enabling the feature maps to obtain more spectral information during the feature extraction process.

In Table 6, the MSAC-Net using 3D convolution shows a 2~7% improvement regarding most indicators.

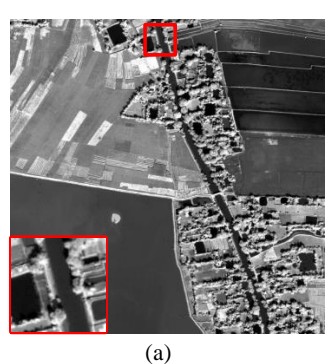 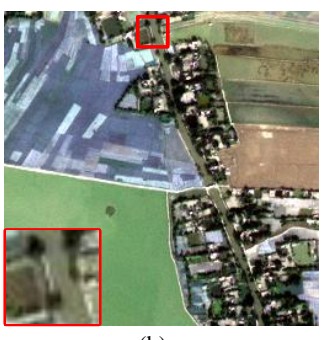 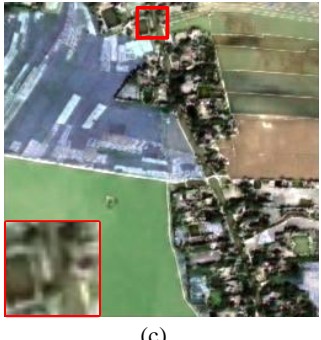 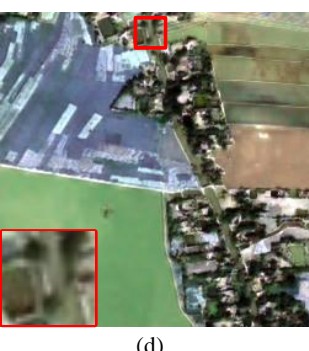

(a)          (b)          (c)          (d)

**Figure 8.** The comparison of 2D and 3D convolution results on the QuickBird dataset. (**a**) PAN. (**b**) The ground truth. (**c**) The 2D convolutional method. (**d**) The 3D convolutional method.

**Table 6.** The comparison of 2D and 3D convolution results on QuickBird.

| Kernel | CC | PSNR | SAM | ERGAS | RMSE | SSIM |
|---|---|---|---|---|---|---|
| 2D | 0.8660 | 19.5025 | 5.4445 | 6.1638 | 0.1059 | 0.6628 |
| 3D | **0.8870** | **20.2168** | **5.1269** | **5.7446** | **0.0975** | **0.7107** |

**Bold** means the best results.

### 4.4. Comparison with the State-of-the-Art Methods

This section compares the proposed method to the eight state-of-the-art methods on the simulated and real dataset. Due to the lack of LR- and HR-MS image pairs, the spatial resolution of both PAN and MS images was reduced four times for training and testing, in accordance with Wald's protocol [51]. During the verification phase, the original MS image and PAN image were used as input and were compared with the whole image and the partial image visually.

#### 4.4.1. Experiments on the Simulated Dataset

The IKONOS and QuickBird simulation data and results are presented in Figures 9 and 10. In Figures 9d–l and 10d–l, the fusion results obtained with GS, Indusion, SR, PNN, PanNet, MSDCNN, MIPSM GTP-PNet and the proposed MSAC-Net method are presented, respectively. Figures 9a–c and 10a–c are LR-MS, PAN and ground truth, respectively. As seen in Figure 9d–h suffer from significant spectral distortions. In contrast, the proposed

method yields a result quite similar to the reference image in terms of visual perception, especially regarding the spatial details. In Figure 10, the resultant images of (d), (e), (j), and (k) are too sharp compared with the reference image, while (h) has a color deviation and (i) is blurred.

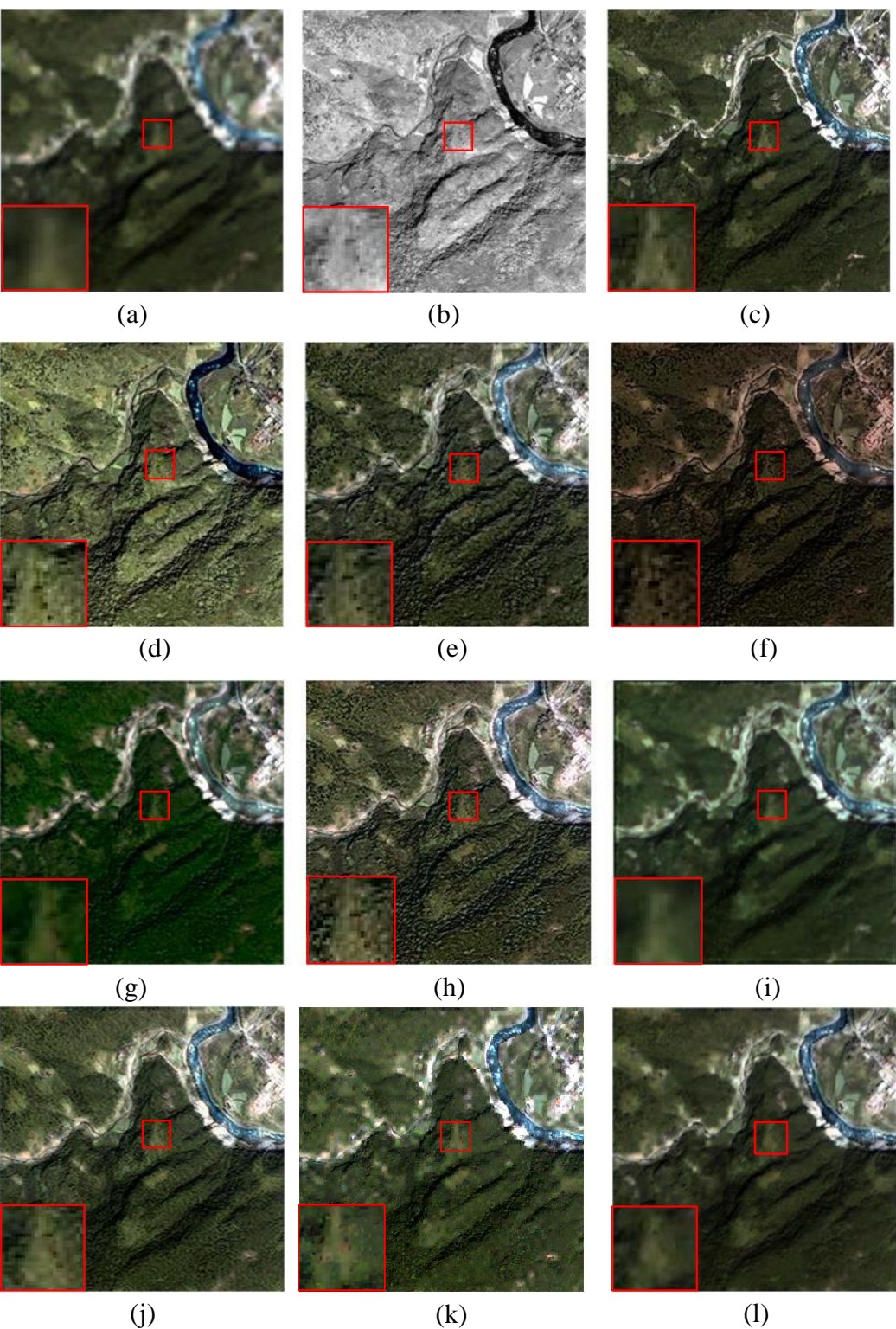

**Figure 9.** The fused results on the simulated IKONOS dataset. (**a**) LR-MS. (**b**) PAN. (**c**) The ground truth. (**d**) GS. (**e**) Indusion. (**f**) SR. (**g**) PNN. (**h**) PanNet. (**i**) MSDCNN. (**j**) MIPSM. (**k**) GTP-PNet. (**l**) MSAC-Net.

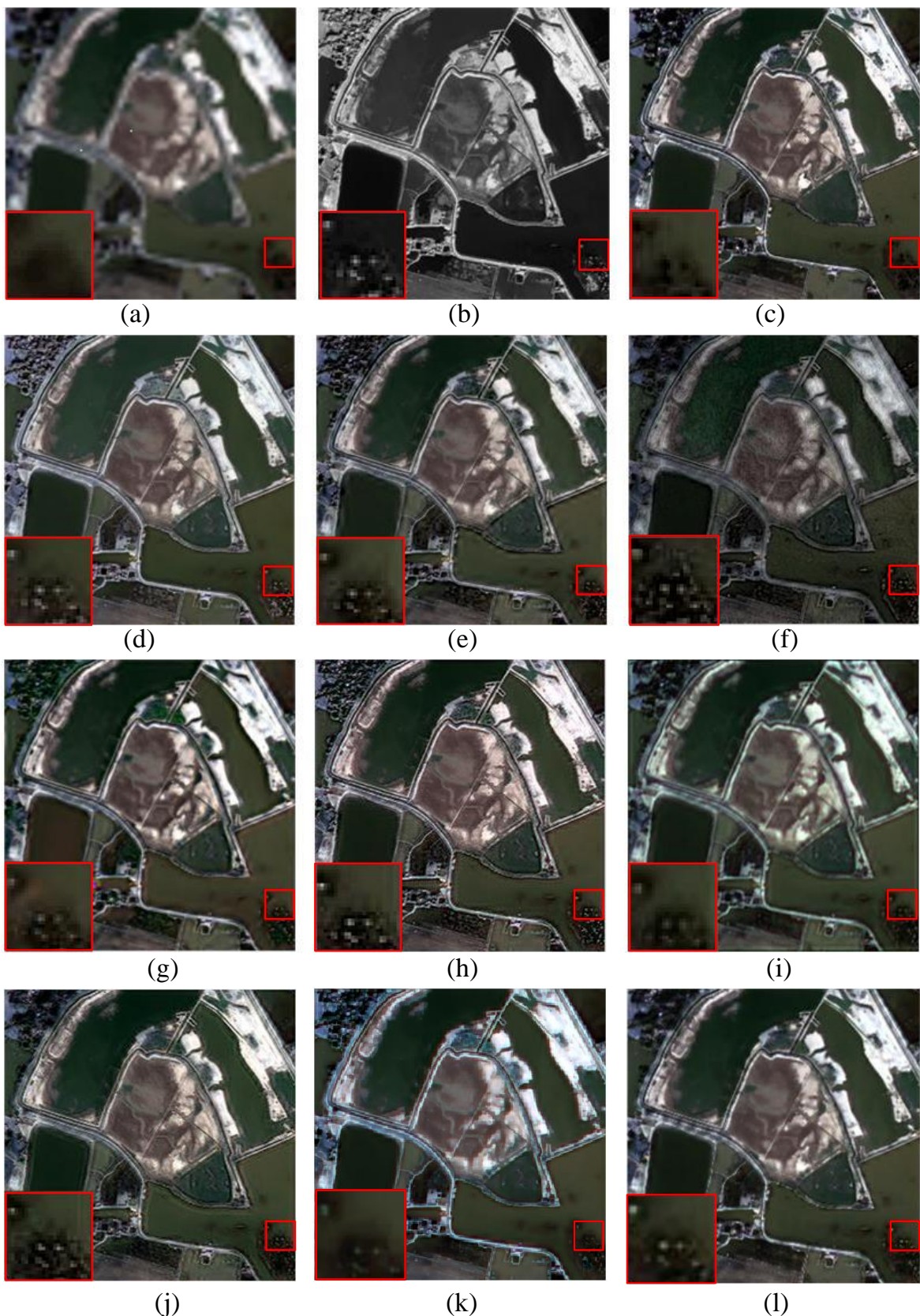

**Figure 10.** The fused results on the simulated QuickBird dataset. (**a**) LR-MS. (**b**) PAN. (**c**) The ground truth. (**d**) GS. (**e**) Indusion. (**f**) SR. (**g**) PNN. (**h**) PanNet. (**i**) MSDCNN. (**j**) MIPSM. (**k**) GTP-PNet. (**l**) MSAC-Net.

In Tables 7 and 8, the methods' results on IKONOS and the QuickBird dataset are listed. The proposed method achieves competitive results on both datasets. With respect to the best values obtained by other methods (as reported in Table 7), although MSAC-Net does not perform well in IKONOS, it is superior only in ERGAS and SSIM. However, other indicators of MSAC-Net in Table 7 are still competitive. With the exception of SAM, Table 8 shows that the proposed method's indicators are superior to those of the current state-of-the-art methods. The results establish the proposed method as satisfactory.

**Table 7.** Quality metrics of the different methods on the simulated IKONOS image.

| Methods | CC | PSNR | SAM | ERGAS | RMSE | SSIM |
|---|---|---|---|---|---|---|
| GS | 0.5320 | 14.3940 | 5.7520 | 20.5527 | 0.1948 | 0.2964 |
| Indusion | 0.8330 | 19.9235 | 6.2750 | 11.3609 | 0.1004 | 0.4136 |
| SR | 0.8206 | 18.8363 | 12.2147 | 15.6579 | 0.1398 | 0.4513 |
| PNN | 0.8952 | 17.7485 | 24.1477 | 12.2516 | 0.0965 | 0.5553 |
| PanNet | 0.8514 | 18.7474 | 6.8293 | 12.6693 | 0.1128 | 0.3931 |
| MSDCNN | 0.9063 | 21.0595 | 5.6025 | 9.8053 | **0.0758** | 0.5728 |
| MIPSM | 0.9070 | 21.0746 | **4.4221** | 9.7840 | 0.0896 | 0.5855 |
| GTP-PNet | **0.9096** | **23.4467** | 6.0760 | 8.1888 | 0.0763 | 0.6020 |
| MSAC-Net | 0.8968 | 21.6184 | 4.4281 | **7.4303** | 0.0830 | **0.6755** |

**Bold** means the best results.

**Table 8.** Quality metrics of the different methods on the simulated QuickBird image.

| Motheds | CC | PSNR | SAM | ERGAS | RMSE | SSIM |
|---|---|---|---|---|---|---|
| GS | 0.8866 | 19.6392 | 2.9178 | 9.5928 | 0.1031 | 0.6973 |
| Indusion | 0.9178 | 21.6106 | **2.6969** | 7.5648 | 0.0822 | 0.7116 |
| SR | 0.9115 | 18.3991 | 3.5598 | 10.7052 | 0.1179 | 0.5897 |
| PNN | 0.9297 | 21.5744 | 5.7713 | 7.2483 | 0.0778 | 0.7398 |
| PanNet | 0.9141 | 21.0793 | 3.8142 | 8.0859 | 0.0883 | 0.7010 |
| MSDCNN | 0.9269 | 21.9366 | 3.8053 | 7.2514 | 0.0794 | 0.7330 |
| MIPSM | 0.9403 | 22.7724 | 4.3359 | 6.6432 | 0.0724 | 0.7300 |
| GTP-PNet | 0.9311 | 21.9180 | 7.9021 | 7.8661 | 0.0962 | 0.7109 |
| MSAC-Net | **0.9515** | **23.0800** | 5.3120 | **6.2297** | **0.0701** | **0.7966** |

**Bold** means the best results.

### 4.4.2. Experiments on the Real Dataset

The results obtained using the real datasets IKONOS and QuickBird are presented in Figures 11 and 12. In Figures 11a–i and 12a–i, the fusion results obtained with GS, Indusion, SR, PNN, PanNet, MSDCNN, MIPSM, GTP-PNet and the proposed MSAC-Net method are presented, respectively. Figures 11j and 12j are the reference PAN images. As seen in Figure 11, the images generated using (a), (b), (e), and (g) are too sharp at the "river" edge. Thus, Figure 11h has an error in the "river" edge. Moreover, the synthetic chroma of Figure 11d deviates, and (f) is blurred. On the other hand, MSAC-Net enhances the spectral information and refines the spatial details.

In Tables 9 and 10, the QNR, $D_s$ and $D_\lambda$ of the eight compared methods are listed for the real IKONOS and QuickBird dataset. The proposed method outperforms other methods regarding all indicators. The indicators in both datasets shown demonstrate that the proposed method can enhances detail extraction and reduces spectral distortion.

Overall, the experiments show the proposed method's capability to retain spectral and spatial information from the original images in the real dataset.

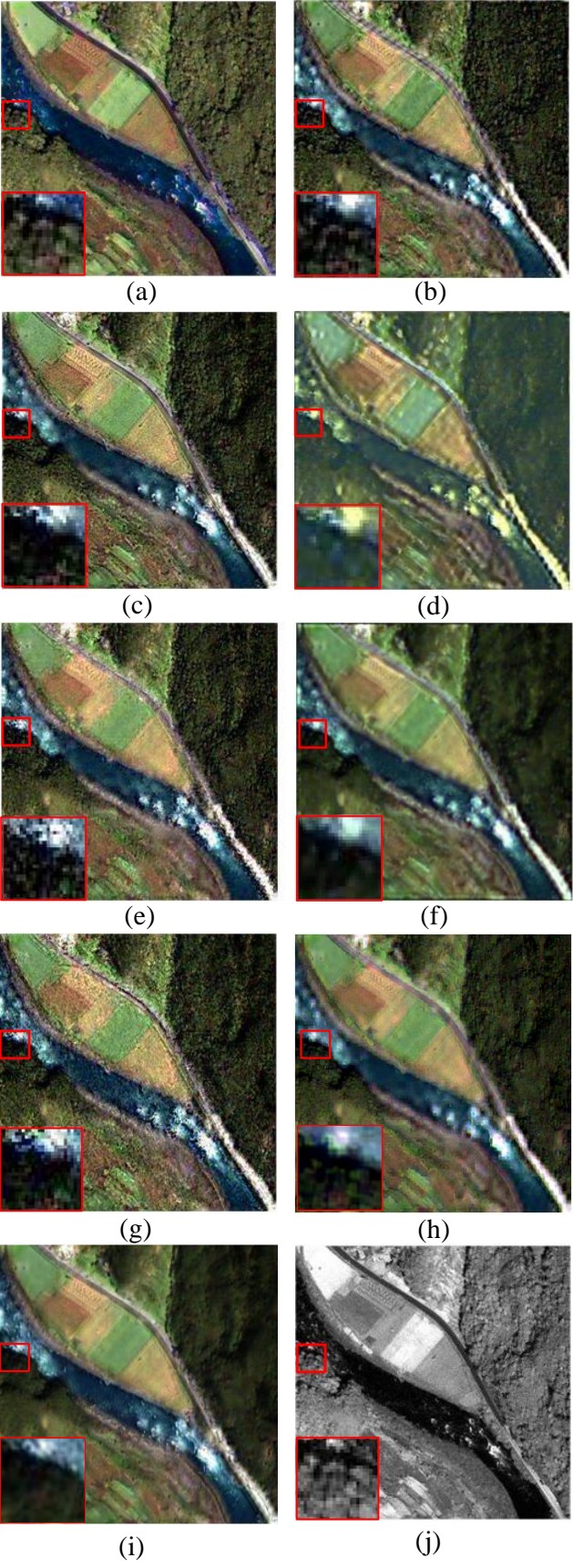

**Figure 11.** The real results on the IKONOS dataset. (**a**) GS. (**b**) Indusion. (**c**) SR. (**d**) PNN. (**e**) PanNet. (**f**) MSDCNN. (**g**) MIPSM. (**h**) GTP-PNet. (**i**) MSAC-Net. (**j**) PAN.

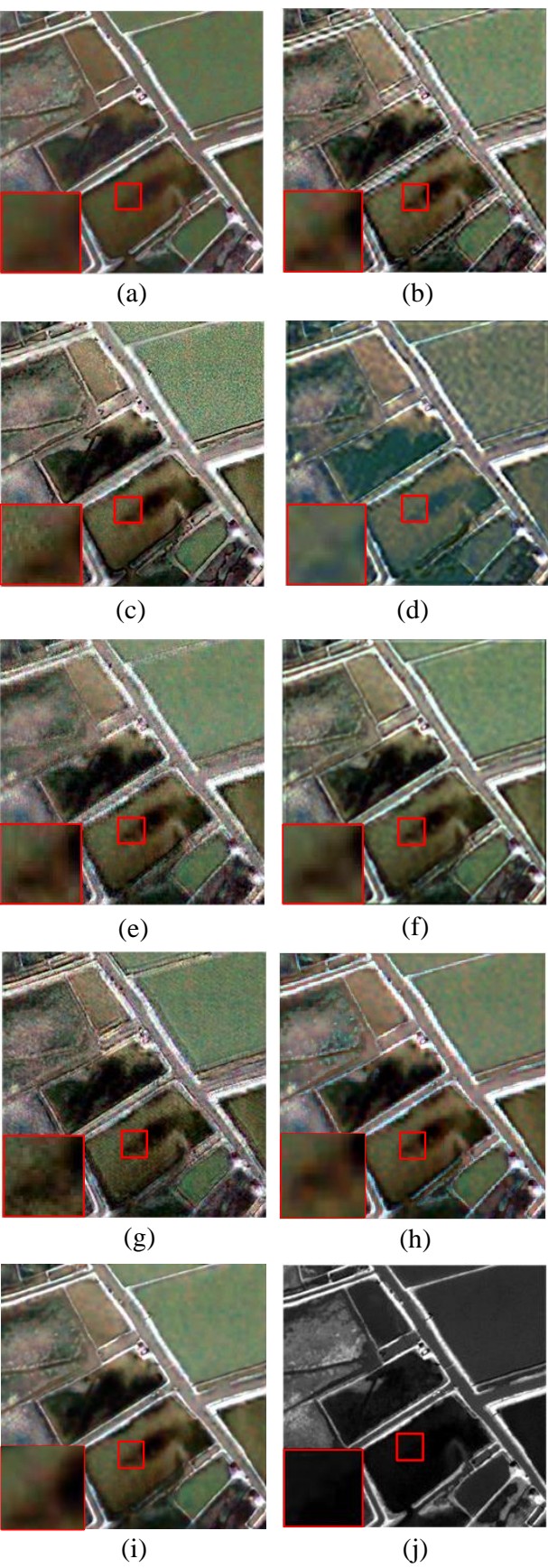

**Figure 12.** The real results on the QuickBird dataset. (**a**) GS. (**b**) Indusion. (**c**) SR. (**d**) PNN. (**e**) PanNet. (**f**) MSDCNN. (**g**) MIPSM. (**h**) GTP-PNet. (**i**) MSAC-Net. (**j**) PAN.

**Table 9.** Quality metrics of the different methods on the real IKONOS dataset.

| Methods | QNR | $D_s$ | $D_\lambda$ |
|---|---|---|---|
| GS | 0.2479 | 0.7366 | 0.0588 |
| Indusion | 0.6785 | 0.3150 | 0.0096 |
| SR | 0.5923 | 0.3934 | 0.0236 |
| PNN | 0.7443 | 0.1579 | 0.1162 |
| PanNet | 0.9251 | 0.0679 | **0.0075** |
| MSDCNN | 0.8016 | 0.1493 | 0.0577 |
| MIPSM | 0.9502 | 0.0267 | 0.0237 |
| GTP-PNet | 0.9494 | 0.0203 | 0.0309 |
| MSAC-Net | **0.9529** | **0.0194** | 0.0282 |

**Bold** means the best results.

**Table 10.** Quality metrics of the different methods on the real QuickBird dataset.

| Methods | QNR | $D_s$ | $D_\lambda$ |
|---|---|---|---|
| GS | 0.3689 | 0.4446 | 0.3358 |
| Indusion | 0.6622 | 0.1870 | 0.1854 |
| SR | 0.7468 | 0.2250 | 0.0364 |
| PNN | 0.7139 | 0.1329 | 0.1767 |
| PanNet | 0.7925 | 0.0469 | 0.1685 |
| MSDCNN | 0.7184 | 0.2098 | 0.0908 |
| MIPSM | 0.8816 | 0.0420 | 0.0797 |
| GTP-PNet | 0.8065 | **0.0141** | 0.1820 |
| MSAC-Net | **0.9546** | 0.0267 | **0.0192** |

**Bold** means the best results.

## 5. Discussion

### 5.1. The Effect of Convolution Times

The number of convolution in the convolution block will be discussed here. According to experience, the higher the number of convolution times, the richer the feature expression of the network should be. However, as shown in Figure 13a, when the number of convolution starts from 2 and keeps increasing, PSNR gradually decreases. This may be due to the excessive times of convolution, which makes the network pay too much attention to the form of the samples, ignoring the characteristics of the data, resulting in over-fitting of the network. Therefore, two $3 \times 3 \times 3$ kernels are used for in training.

### 5.2. The Effect of Multi-Scale Information Weight $\lambda$

This subsection discusses the effect of weight $\lambda$ in Equation (10). Figure 13b shows that the increase in weight $\lambda$ from 0.01 to 1 gradually strengthens the constraints on scale information in MSAC-Net and the resultant feature maps comprise sufficient details at each scale to enhance the results. When $\lambda$ increases to 10, the network pays more attention to the scale information than the fusion result, affecting the MSAC-Net's final result. Therefore, to balance the scale information and fusion result, $\lambda$ is set as to 1.

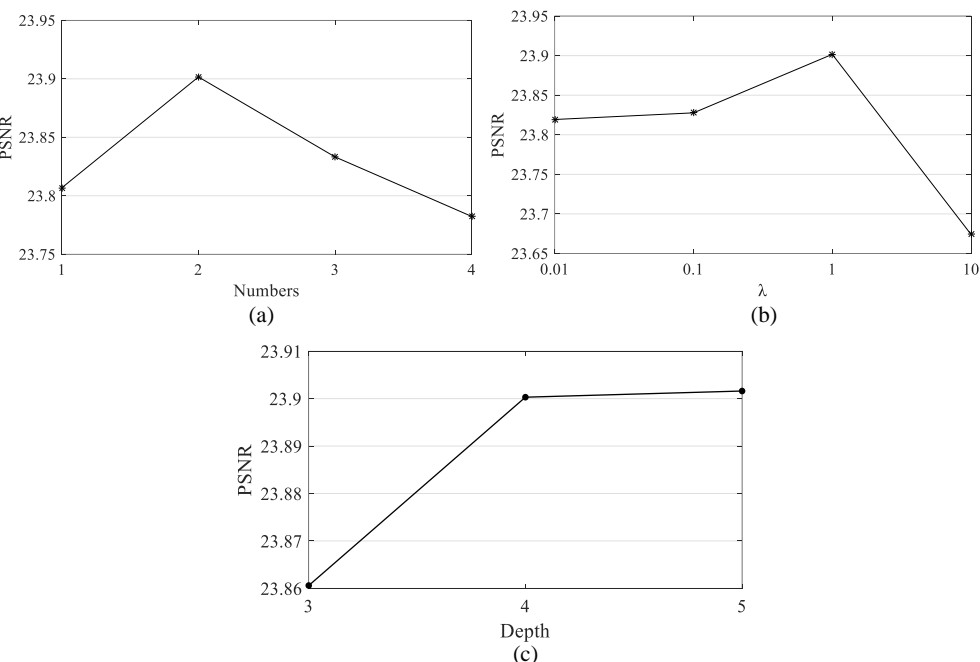

**Figure 13.** Discuss the effects of parameters (**a**) The effect of convolution times. (**b**) The effect of multi-scale information weight $\lambda$. (**c**) The effect of network depth.

### 5.3. The Effect of Network Depth

A final parameter effect that needs to be investigated is the influence of the MSAC-Net's depth.

As is known to all, increasing the network depth can improve the effect of feature extraction, but a too deep network will lead to degradation. Therefore, based on experience, we discuss the depth of MSAC-Net between three and five layers. Figure 13c shows that as the network depth increases, the average PSNR of the MSAC-Net on the test set gradually improves. Therefore, according to the results in Figure 13c, the depth of MSAC-Net is set to five layers.

Following these experiments, the weight $\lambda$ was set to 1, the convolution times was 2 and the depth was 5 in MASC-Net.

## 6. Conclusions

This work introduces a novel 3D multi-scale attention deep convolutional network (MSAC-Net) method for MS imagery pansharpening. The proposed MSAC-Net method utilizes a 3D deep convolutional network appropriate for the MS images' characteristics. Moreover, the method integrates the attention mechanism and deep supervision mechanism for spectral and spatial information preservation and extraction. The conducted experiments show that MSAC-Net using 3D convolution achieves better quantitative and visual pansharpening performance than the network with 2D convolution. Exhaustive experiments investigated and analyzed the effects of the designed attention module, multi-scale cost function and three critical MSAC-Net factors. The experimental results demonstrate that every designed module positively affects spatial and spectral information extraction, enabling MSAC-Net to achieve the best performance in the appropriate parameters' range. Compared to the state-of-the-art pansharpening methods, the proposed MSAC-Net achieved comparable or even superior performance on the real IKONOS and QuickBird satellites' datasets. Building on the results reported in this study, one can conclude that MSAC-Net is a promising multi-spectral imagery pansharpening method. In future work, we will explore how to make more efficient use of the attention mechanism to obtain smaller CNN while ensuring the quality of the fusion image.

**Author Contributions:** E.Z.: Conceptualization, Writing—review and editing, Supervision. Y.F.: Methodology, Software, Formal analysis, Writing—original draft. J.W.: Conceptualization, Writing—review and editing. L.L.: Supervision, Investigation, Data curation. K.Y.: Resources. J.P.: Visualization, Project administration. All authors have read and agreed to the published version of the manuscript.

**Funding:** This work is supported by the National Natural Science Foundation of China (No. 62006188, 62101446), the Xi'an Key Laboratory of Intelligent Perception and Cultural Inheritance (No. 201-9219614SYS011CG033), Key Research and Development Program of Shaanxi (No. 2021ZDLSF06-05), the International Science and Technology Cooperation Research Plan of Shaanxi (No. 2022KW-08), the Program for Chang-jiang Scholars and Innovative Research Team in University (No. IRT_17R87) and the QinChuangyuan high-level innovation and entrepreneurship talent program of Shaanxi (2021QCYRC4-50).

**Data Availability Statement:** Not applicable.

**Conflicts of Interest:** The authors declare no conflict of interest.

**Abbreviations**

| | |
|---|---|
| MS | Multi-spectral |
| PAN | Panchromatic |
| HR | High resolution |
| LR | Low resolution |
| CNN | Convolutional neural network |
| PNN | Pansharpening neural network |
| MSDCNN | Multi-scale and multi-depth CNN |
| AG | Attention gate |
| GS | Gram–Schmidt |
| CC | Correlation coefficient |
| PSNR | Peak signal-to-noise ratio |
| SAM | Spectral angle mapper |
| RMSE | Root mean square error |
| ERGAS | *Erreur relative globale adimensionnelle de synthèse* |
| SSIM | Structural similarity index measurement |
| QNR | Quality without reference |
| Q | Image quality index |

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
