# Peer review of "MSAC-Net: 3D Multi-Scale Attention Convolutional Network for Multi-Spectral Imagery Pansharpening"

_remotesensing, doi:10.3390/rs14122761_

Round 1

Reviewer 1 Report

See attached file, please.

Reviewer 2 Report

This paper proposes a new pansharpening technique based on deep learning methods. The paper is well written and proposes a mechanism that may be helpful in other remote sensing studies. I recommend the publication with minor reviews.

My main suggestions are listed below:
•    Figure 4 is not called in the text. I believe it would be better suited if the location was in the 4.1.1 subsection. This figure also needs labels (e.g., A, B, C..). Remember, the figures must be self-explanatory. The authors could call the figure right here: “To test the effectiveness of MSAC-Net, we used data sets collected by IKONOS and 163 QuickBird satellites (Figure 4)”. Please check if any other tables or figures were not referenced in the text.
•    The paper has a vast number of abbreviations and acronyms. I suggest the authors make a table with all acronyms and abbreviations. In some cases, I had a hard time finding some of them. I believe the readers will also have this problem.
•    I believe the authors meant Blue instead of Blur in Table 2. A quick note to the authors, all figures and table legends should explain the acronyms and abbreviations.
•    Some phrases are too short, and there is no need to start a new paragraph in some cases. For example, in lines 195, 210, 233, 275. Please check the entire text. Section 4.5.3. is full of very short lines, which is not necessary.
•    The authors need to say that MSE() refers to the Mean squared error in equation 12. Please carefully check all the equations to see if any other information is missing.
•    Since the authors adopted a style of writing that uses many tables. It would be informative to have a table with the dataset image settings. Image dimension, number of patches.
•    The authors use a validation set. Did the authors save the best model in the validation set? This is a good habit that I believe the authors did, but it should be written in the text.
•    Since the authors are proposing a new method, I believe it is crucial to insert a nice and informative GitHub so that other researchers can easily replicate the experiments.
•    I believe this work could improve by introducing a discussion section to present the novelties and future applications more evidently. Some suggestions: (1) explain how this work advances in this field compared to other results, (2) Explain possible limitations, (3) explain possible applications, (4) how do you reconstruct an entire scene? is it using sliding windows? how does it differ from sliding windows in segmentation problems that often use the U-Net? and (5) some deep learning studies use more spectral bands to achieve better classification, object detection, and segmentation results. Some works even compare using more spectral bands. The authors could analyze this topic and explain how the proposed method could provide better results.

Reviewer 3 Report

The authors present a deep learning approach based on a U-Net convolutional neural network for Multi-spectral Imagery Pansharpening. My expertise in this context lies in the application of neural networks for geophysical problems. In general, I think that this study presents a useful step forward for the community and that it builds on the present state of knowledge. I have a couple of concerns with the presented methodology that should be addressed. The manuscript should be returned to the authors for a round of major revisions and re-evaluated afterward. However, more details about the model development and optimization need to be clarified. My major comments and questions are as follows:

  • What is the uniqueness of the proposed algorithm and its potential impacts, over other recently established states of the art Res-Net based Conventional convolutional neural networks (CNNs) based semantic segmentation methods such as Mask RCN for Semantic Segmentation of Remote Sensing Images (Witharana, et al.2021; Zhang et al. 2020) in remote sensing applications? Please introduce these works and their potential impact. The authors should explain this aspect in the introduction section. Otherwise, the readers cannot see the importance or uniqueness of your proposed methods over other techniques.

Zhang et al. 2020, “Transferability of the Deep Learning Mask R-CNN Model for Automated Mapping of Ice-Wedge Polygons in High-Resolution”. Remote Sens 2018, 10, 1487

Witharana, Chandi, et al. "Understanding the synergies of deep learning and data fusion of multispectral and panchromatic high resolution commercial satellite imagery for automated ice-wedge polygon detection." ISPRS Journal of Photogrammetry and Remote Sensing 170 (2020): 174-191.

  • Can you explain the high-resolution features and high-level semantic information with examples?
  • What Optimizer, loss function did you use? How did you choose essential tuned hyperparameters (e.g. number of hidden nodes, learning rate, etc.)? You should provide a table for hyperparameters settings. What is the fundamental theory about this information? What software or programs and HPC resources are used to fit the DL model?
  • How do the authors come up with the current optimized DLNN structure? How did you create DL optimized model without showing any fundamental results? Although you discussed a lot about loss functions still missing a few required results for model optimization. Several plots are mandatory to justify the model optimization (i)epoch vs loss (ii) Epoch vs accuracy 
  • Can you please provide a table for validation/training/testing samples along with other information such as sample number, patch, sample source, training and validation sites, repository etc?
  • In the discussion section, you should discuss more your results vs previous research.
  • Can you explain more about overfitting issues?

Round 2

Reviewer 1 Report

1. This version of the manuscript has been substantially revised to make it easier to understand, especially the notation and architectural analysis of the proposed method.

2. For my suggestions, the author has corrected and explained most of them in this version.

3. Neither of the two algorithms in my comment that the authors would like to compare in this edition of the article are compared, but adding another algorithm, unfortunately but barely acceptable. Also gives us a better idea of the effect of performance of the proposed method.

Reviewer 3 Report

The authors significantly improved the quality of the paper by addressing most of the previous comments. This research work will be very effective for the remote sensing and AI  community. I recommend the manuscript for publication!